# Unveiling the Secrets of Calcium-Dependent Proteins in Plant Growth-Promoting Rhizobacteria: An Abundance of Discoveries Awaits

**DOI:** 10.3390/plants12193398

**Published:** 2023-09-26

**Authors:** Betina Cecilia Agaras, Cecilia Eugenia María Grossi, Rita María Ulloa

**Affiliations:** 1Laboratory of Physiology and Genetics of Plant Probiotic Bacteria (LFGBBP), Centre of Biochemistry and Microbiology of Soils, National University of Quilmes, Bernal B1876BXD, Argentina; 2National Scientific and Technical Research Council (CONICET), Buenos Aires C1425FQB, Argentina; cecigrossi@gmail.com; 3Laboratory of Plant Signal Transduction, Institute of Genetic Engineering and Molecular Biology (INGEBI), National Scientific and Technical Research Council (CONICET), Buenos Aires C1425FQB, Argentina; 4Biochemistry Department, Faculty of Exact and Natural Sciences, University of Buenos Aires (FCEN-UBA), Buenos Aires C1428EGA, Argentina

**Keywords:** calcium, Ca^2+^ channels, Ca^2+^ pumps, CaBP, plant growth-promoting rhizobacteria

## Abstract

The role of Calcium ions (Ca^2+^) is extensively documented and comprehensively understood in eukaryotic organisms. Nevertheless, emerging insights, primarily derived from studies on human pathogenic bacteria, suggest that this ion also plays a pivotal role in prokaryotes. In this review, our primary focus will be on unraveling the intricate Ca^2+^ toolkit within prokaryotic organisms, with particular emphasis on its implications for plant growth-promoting rhizobacteria (PGPR). We undertook an in silico exploration to pinpoint and identify some of the proteins described in the existing literature, including prokaryotic Ca^2+^ channels, pumps, and exchangers that are responsible for regulating intracellular Calcium concentration ([Ca^2+^]_i_), along with the Calcium-binding proteins (CaBPs) that play a pivotal role in sensing and transducing this essential cation. These investigations were conducted in four distinct PGPR strains: *Pseudomonas chlororaphis* subsp. *aurantiaca* SMMP3, *P. donghuensis* SVBP6, *Pseudomonas* sp. BP01, and *Methylobacterium* sp. 2A, which have been isolated and characterized within our research laboratories. We also present preliminary experimental data to evaluate the influence of exogenous Ca^2+^ concentrations ([Ca^2+^]_ex_) on the growth dynamics of these strains.

## 1. Introduction

Calcium (Ca^2+^) is the fifth most abundant element of the Earth’s crust (constituting up to 3%) and is present in fresh and saltwater at concentrations ranging from micro to millimolar. Some paleontologists propose that 3.5 billion years ago, when life began, the concentration of free Ca^2+^ surrounding the first cells was somewhere in the range of 100 nM [1]. Hence, the very first cells acquired a very low Ca^2+^ content in their cytoplasm and lived in a low Ca^2+^ environment (nanomolar range). Low Ca^2+^ in the cytosol of primeval cells is also compatible with energetics based around ATP and the usage of DNA/RNA for genetic encoding [2]. A prolonged elevation of intracellular free [Ca^2+^] can lead to cell death since it will irreversibly damage mitochondria and can cause chromatin condensation, precipitation of phosphate, protein aggregation, and disruption of lipid membranes through the activation of proteases, nucleases, and phospholipases [3,4].

In the biological world, Ca^2+^ is present in three forms: free, bound to proteins, and trapped in mineral form, mainly with phosphate as hydroxyapatite. Free Ca^2+^ concentration [Ca^2+^] in plant cells varies in very wide limits depending on its location. Under resting conditions in the cytoplasm, it is ~10^−7^ M, while in intracellular compartments, such as the endoplasmic reticulum (ER), mitochondria, chloroplasts, or the vacuole, [Ca^2+^] is 1–5 × 10^−4^ M. These organelles that can accumulate Ca^2+^ are referred to as Ca^2+^ stores. Extracellular [Ca^2+^] in the cell wall or apoplast is ~10^−3^ M [5]. The intracellular free [Ca^2+^] in prokaryotes is tightly regulated in the 100 to 300 nM range [6,7]. In contrast, the free Mg^2+^ concentration is high (0.5–1 mM) and practically constant. The Ca^2+^ ion can accommodate 4–12 negatively charged oxygen atoms in its primary coordination sphere, with 6–8 being the most common. It can form stable complexes with bulky and irregularly shaped anions (proteins) since its coordination sphere is rather flexible. Ca^2+^ binds water much less tightly than Mg^2+^ and precipitates phosphate. Mg^2+^ ions compete for most Ca^2+^ binding sites; however, proteins bind Mg^2+^ ions 4 to 5 orders of magnitude weaker [8].

Ca^2+^ was selected early in evolution as a signaling molecule to be used by both prokaryotes and eukaryotes. It was reported to be involved in the regulation of cell division, chemotaxis, and cell differentiation processes in prokaryotes (revised in [9]) and is a well-established second messenger in eukaryotes that mediates cell responses to different environmental cues. Upon endogenous or exogenous stimuli, cells trigger short-lived and often highly localized Ca^2+^ transients. It is surprising how changes in intracellular Ca^2+^ concentration [Ca^2+^] can regulate cellular functions in such a multitude of ways with distinct spatiotemporal outcomes. A rise in cytosolic free Ca^2+^ is responsible for initiating cellular events such as movement, secretion, transformation, and division. Ca^2+^ transients are named sparks, embers, quarks, puffs, blips, or waves, depending on their exact spatial and temporal properties and the cell type in which they occur. Furthermore, the Ca^2+^ signals can be highly repetitive, forming Ca^2+^ oscillations (revised in [10]). The intracellular [Ca^2+^] transients result from a complex interplay between Ca^2+^ influx/extrusion systems, mobile/stationary Calcium binding proteins (CaBPs), and intracellular sequestering mechanisms. Underlying the speed and effectiveness of Ca^2+^ is the 20,000-fold gradient maintained by cells between their intracellular (~100 nM free) and extracellular (1–5 mM) concentrations. Ca^2+^ is then rapidly excluded from the cytosol. Cells invest much of their energy to maintain the basal [Ca^2+^]; unlike other complex molecules, Ca^2+^ cannot be chemically altered; thus, cells must chelate, compartmentalize, or extrude it. Eukaryotic and prokaryotic cells employ Ca^2+^ pumps and Ca^2+^ exchangers to keep their basal concentration at a very low (~20–200 nM) level.

These Ca^2+^ signatures (differing in amplitude, frequency, and localization) are decoded byCaBPs. Hundreds of proteins harbor Ca^2+^ binding domains either to buffer or lower Ca^2+^ levels or to trigger cellular processes. A variety of proteins ranging from calmodulin to membrane channel voltage sensors or nuclear DNA-binding proteins present EF hand domains (Nakayama and Kretsinger, 1994) that consist of a ~12 amino acid loop between two orthogonal α helices. The affinities of EF hand domains for Ca^2+^ vary ~100,000-fold depending on a variety of factors ranging from critical amino acids in the Ca2+ binding loop to sidechain packing in the protein core [11].

CaBPs are the fastest players within the Ca^2+^ toolkit, responding within microseconds to [Ca^2+^] changes. The CaBPs compete for Ca^2+^ and play a direct role in modulating Ca^2+^ transients and the resulting biochemical message. Upon Ca^2+^ binding, these proteins undergo a conformational change and are able to interact with the effector proteins that amplify the signal, allowing the cell to respond to the stimuli. Ca^2+^ can also play a role in regulating lipid synthesis and, consequently, membrane composition. It has been indicated that Ca^2+^ is involved in nucleoid structure and the function of proteins required for DNA replication contributes to the concept.

The role of Ca^2+^ is very well studied and understood in eukaryotes, but the current knowledge, mostly based on human pathogenic bacteria, indicates that this ion is also a key player in prokaryotes. In this review, we will very briefly summarize the multifaceted role of Ca^2+^ in plant-microorganism interactions, and we will focus on the Ca^2+^ toolkit in prokaryotes, particularly in plant growth-promoting rhizobacteria (PGPR). Additionally, we determined the effect of exogenous Ca^2+^ concentrations on the growth of four PGPR strains that were isolated and characterized in our labs [12,13,14,15], and we conducted an in silico search to identify some of the prokaryotic CaBPs described in the literature for these isolates.

## 2. Calcium Toolkit in Plants: Its Role during Biotic Interactions

Ca^2+^ is an essential plant nutrient. Its deficiency is rare in nature, but Ca^2+^-deficiency disorders occur in horticulture. On the other hand, excessive Ca^2+^ on calcareous soils restricts plant communities. This cation is taken up by roots from the soil and is translocated to the shoot via the xylem; however, its movement must be finely tuned to allow root cells to signal using [Ca^2+^], control the rate of Ca^2+^ delivery to the xylem, and prevent the accumulation of toxic cations in the shoot [16]. Plants have developed a complex Ca^2+^ toolkit that includes tightly regulated proteins responsible for Ca^2+^ influx and efflux, and those that decode and transduce the Ca^2+^ oscillations. Different types of Ca^2+^ channels, ATP-driven Ca^2+^ pumps, and Ca^2+^ exchangers localized in the plasma membrane, in the ER, chloroplasts, and vacuole contribute to Ca^2+^ homeostasis in Arabidopsis cells. Excellent reviews describe the families of ion channels including cyclic nucleotide-gated channels (CNGCs), ionotropic glutamate receptors (GLRs), the slow vacuolar two-pore channel 1 (TPC1), voltage-dependent hyperpolarization-activated Ca^2+^ permeable channels (HACCs), annexins, and several types of mechanosensitive channels that mediate Ca^2+^ influx in plant cells. On the other hand, Ca^2+^ efflux is regulated by Calcium exchangers such as CAX, EFCAX, CCX, magnesium/proton exchangers (MHX), which are plant homologs of animal Na^+^/Ca^2+^ exchangers, and Na^+^/Ca^2+^-K^+^ exchangers and Ca^2+^ ATPases such as ACAs and 2 ECAs (revised in [17,18]).

The different Ca^2+^ signatures generated by external and internal stimuli are decoded by Calcium sensors, including Calcium-binding proteins such as Calmodulins, and calcineurin B-like (CBL) proteins, which in turn modulate protein phosphorylation through the activation of calmodulin-dependent protein kinases (CaMK) and CBL-interacting protein kinases (CIPKs), or by Calcium sensor-transducers that include Calcium-dependent protein kinases (CDPKs) and CDPK-related kinases (CRKs) [19].

Plant-microorganism interactions play a crucial role in shaping plant health, development, and defense against pathogens. Plants have developed several strategies to promote or stimulate the growth of particular microorganisms in their rhizosphere [20], mainly based on their root exudates [21,22,23], thus shaping the plant rhizobiome [24]. Among the most studied PGPRs with beneficial effects on plant health are different species of *Pseudomonas, Azospirillum, Bacillus*, *Burkholderia*, *Methylobacterium*, *Trichoderma*, and actinomycetes [25,26]. Plant growth promotion may occur by several mechanisms: while biofertilization and phytostimulation have a direct effect on plant development, biocontrol contributes to plant health by disturbing pathogen growth [27]. There are several biocontrol strategies: direct antagonism, elicitation of the induced systemic resistance (ISR) of plants, niche and nutrients competition, parasitism, or quorum quenching [27,28].

Ca^2+^ is involved in the interaction of plants with pathogenic and benefic microorganisms, as well as in the response to abiotic stresses such as salt, drought, heat, or cold (revised in [29]). Different Calcium signatures were identified using luminescent and fluorescent-based Calcium-imaging techniques; among them are the stimulus-specific oscillations reported in guard cells during stomata closure or those associated with symbiosis signaling triggered by nod factors from rhizobia or mycorrhizal fungi [30,31,32,33]. The dynamic interplay between plants and microorganisms involves intricate Calcium-mediated signaling pathways that determine the outcome of these interactions. Upon recognition of pathogen-associated molecular patterns (PAMPs) or symbiotic signals, plants trigger an influx of Calcium ions into the cytoplasm from intracellular stores and/or the extracellular space [34,35]. This Calcium signature serves as an essential trigger for initiating defense responses against pathogens or establishing symbiotic associations. An update of the state of the art on plant biotic interactions has been recently compiled in a theme issue on Biotic interactions edited by [36].

In pathogenic interactions, elevated cytosolic Ca^2+^ levels lead to the activation of various downstream signaling components, including protein kinases, transcription factors, and reactive oxygen species (ROS) production. These signaling cascades culminate in the expression of defense-related genes, the synthesis of antimicrobial compounds, reinforcement of the cell wall, and programmed cell death to limit pathogen spread. These mechanisms are part of the Systemic Acquired Resistance (SAR). During SAR, pathogen-induced Ca^2+^ waves propagate systemically, transmitting the priming signal to distant plant tissues and inducing defense responses. This systemic Ca^2+^ signaling enables the plant to respond more effectively to subsequent pathogen attacks. On the other hand, effector-triggered immunity (ETI) is a robust defense mechanism activated by plants in response to specific pathogen effectors. Ca^2+^ signaling contributes to the hypersensitive response (HR), a form of programmed cell death localized at the infection site. Ca^2+^ influx triggers a cascade of events, leading to the activation of defense genes and hypersensitive cell death, limiting the spread of pathogens (revised in [37,38,39]). Ca^2+^ has also been implicated in defense priming, which involves the pre-activation of defense responses in plants to enhance their ability to combat future pathogen attacks. Priming is associated with the accumulation of Ca^2+^ in specific cellular compartments, which enables a faster and stronger defense response upon subsequent pathogen encounters.

Microorganisms have evolved diverse strategies to manipulate host Ca^2+^ signaling for their own benefit. Pathogens can secrete effector molecules that interfere with Ca^2+^ signaling pathways, suppressing plant defense responses, and promoting pathogen colonization [35]. Conversely, beneficial microorganisms can elicit Ca^2+^ influx and other signaling events that promote plant growth, nutrient uptake, and overall benefic interactions. Ca^2+^ mediates crosstalk between different signaling pathways involved in plant-microorganism interactions. For example, Ca^2+^ signaling intersects with phytohormone signaling pathways such as salicylic acid (SA), jasmonic acid (JA), and ethylene (ET) [40,41]. This integration allows plants to coordinate defense responses against pathogens and fine-tune symbiotic and PGPR associations.

## 3. Calcium Toolkit in Microorganisms

The importance of Ca^2+^ homeostasis in bacteria has been studied for the last 30 years because it has been challenging to measure the intracellular [Ca^2+^] in such tiny cells [42]. Thus, although in eukaryotes the Ca^2+^ role is extensively defined and described, in prokaryotes its role is still elusive, but there is increasing evidence for Ca^2+^ as a regulator in many bacterial processes, such as sporulation, virulence, septation, chemotaxis, cell wall maintenance, and phosphorylation [42,43,44]. Some Ca^2+^ toolkits have been found in bacterial genomes [45]; these proteins are involved in defining the Ca^2+^ signatures [9].

Bacteria have been shown to maintain low free intracellular [Ca^2+^]_i_ against large changes in external [Ca^2+^]. The evolution of the Ca^2+^-dependent system of the Eukaryotic Endoplasmic Reticulum using sequence-based homology searches was analyzed, and it was found that at least three protein families of the ER Ca^2+^-stores system have a clearly identifiable bacterial provenance, namely the P-type ATPase pump SERCA, sarcalumenin, and calmodulin and related EF-hand proteins [46]. Phylogenetic analyses suggest that the P-type ATPases SERCA and plasma membrane Ca^2+^-transporting ATPase (PMCA) were both present in the last eukaryotic common ancestor and are most closely related to bacterial P-type ATPases [47], which commonly associate with transporters (e.g., Na^+^-Ca^2+^ antiporters), ion exchangers (Na^+^-H^+^ exchangers), permeases, and other distinct P-type ATPases in conserved gene-neighborhoods, suggesting a role in the maintenance of ionic homeostasis even in bacteria.

The maintenance of a low intracellular free Ca^2+^ concentration is required not only to protect the cell from the toxic effects of Ca^2+^ but also to permit the use of Ca^2+^ as a second messenger [43]. The metabolic apparatus that serves this function involves Ca^2+^ channels, Ca^2+^ antiporters, and ATP-dependent Ca^2+^ pumps, together with intracellular Ca^2+^-binding buffers, such as the *Saccharopolyspora erythraea* protein, calerythrin (Figure 1A).

The approach adopted for the identification of Ca^2+^-“associated” proteins in *E. coli* provided a better understanding of the relevance of CaBPs in bacteria. Several CaBPs have been described in prokaryotic microorganisms, using a combination of molecular and bioinformatic approaches. The finding of this kind of protein in prokaryotes was based on sequence similarity with eukaryotic Ca^2+^ binding motifs, such as the EF-hand (Figure 1A) [48], the EF-hand-like (or pseudo-EF-hand; Figure 1B) [49], β-roll (Figure 1C) [50,51,52], the Greek key motif (Figure 1D) [53] and a particular Greek key motif, the Bacterial Immunoglobulin-like (BIg) domain (Figure 1E) [54]. Thus, bacterial proteins are variable, with structural diversity as in eukaryotes. In these proteins, Ca^2+^ binding can act as a stabilizer or effector. As a stabilizer, Ca^2+^ can promote the stabilization of the structure of proteins [55], the folding into a functional state [56,57], or the membrane organization [58]. However, most of these functions were not demonstrated by in vitro assays with the predicted bacterial proteins. Even less, the activity of Ca^2+^-related proteins has been demonstrated in soil microorganisms or those inhabiting ecosystems in interaction with plants.

**Figure 1 plants-12-03398-f001:**
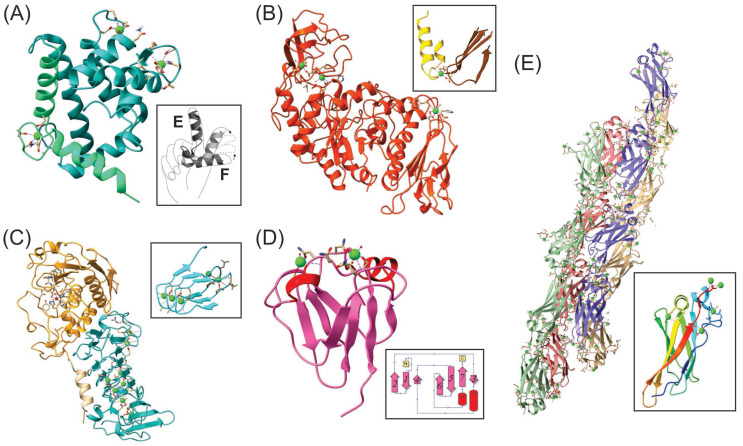
Three-dimensional structures of different bacterial proteins with demonstrated Ca^2+^-binding motifs. In all the structures, coordinating aminoacids are shown and the Ca^2+^ ions are colored in green. The visualizations were performed with ChimeraX version 1.5 (https://www.rbvi.ucsf.edu/chimerax, accessed on 5 September 2023). (**A**) NMR solution structure of calerythrin from *Saccharopolyspora erythraea*, which contains 4 putative EF-hand motifs and 3 Ca^2+^-binding sites. One of the EF-hand motifs with a bound Ca^2+^ is shown close-up in light green. Calerythrin belongs to the family of sarcoplasmic Ca^2+^-binding proteins (SCPs), which have a compact globular structure and function as intracellular Ca^2+^ buffers. To illustrate, we include a symbolic representation of the EF-hand motif, where helix E winds down the index finger and helix F winds up the thumb of a right hand and moves in different conformations (adapted from [59]). The structure was obtained from the Protein Data Bank (https://www.rcsb.org/structure/1NYA, accessed on 5 September 2023). (**B**) X-ray structure of the α-amylase from *Bacillus licheniformis* with an EF-hand-like motif, where three Ca^2+^ ions coordinate. Although the coordination properties remain similar with the canonical helix–loop–helix of an EF-hand motif, this motif contains deviations in the secondary structure of the flanking sequences and/or variation in the length of the Ca^2+^-coordinating loop. One of the EF-hand-like motifs of the amylase, in which a helix is replaced by a β-sheet, is shown in detail. The structure of the amylase was obtained from PDB (https://www.rcsb.org/structure/1BLI, accessed on 5 September 2023). (**C**) X-ray structure of the alkaline protease AprA from *Pseudomonas aeruginosa* PA01. The structure shows the two domains present in the protein. In cyan, the β-roll motif stands out, with 8 Ca^2+^ ions coordinated; a fragment is shown in detail, in which two parallel β-sheets form the motif. This motif is typical of the repeat-in-toxin (RTX) proteins. The structure was obtained from PDB (https://www.rcsb.org/structure/1KAP, accessed on 5 September 2023). (**D**) X-ray structure of the N-terminal domain of Protein S from *Myxococcus xanthus*, with two Greek key motifs in a β-sandwich arrangement that coordinate two Ca^2+^ ions (https://www.rcsb.org/structure/1NPS, accessed on 5 September 2023). The 2D topology is shown in detail (obtained from PDBsum, http://www.ebi.ac.uk/thornton-srv/databases/pdbsum/, accessed on 5 September 2023), where the typical Greek key design is observed, with the seven β chains numerated. (**E**) X-ray crystal structure of the RII tetra-tandemer of the bacterial ice-binding adhesin from *Marinomonas primoryensis*. Each chain of the tetra-tandemer is colored differently (https://www.rcsb.org/structure/4p99, accessed on 5 September 2023). The monomer, shown in detail, contains seven multicolored β-strands and requires five Ca^2+^ for folding (https://www.rcsb.org/structure/4KDV, accessed on 5 September 2023). The native adhesin contains four regions that accomplish different roles. This structure is part of region II (RII), the extender region, which consists of 120 tandem copies of the monomer on each chain. Each repeat folds as a Ca^2+^-bound immunoglobulin (Ig) like (Big) β-sandwich domain, a particular Greek key motif. This comprises 70 to 100 amino acid residues divided into two stacked β-sheets, with a total of at least seven and as many as ten anti-parallel β-strands. In adhesins, Ca^2+^ rigidifies the linker region between each domain. LapA and LapF share the structural basis for adhesins with this protein [60], hence Ca^2+^-induced rigidity of tandem Ig-like repeats in large adhesins might be a general mechanism used by bacteria to bind to their substrates and help colonize specific niches.

## 4. Calcium Signaling in Symbiotic Associations

Ca^2+^ plays a crucial role in establishing and maintaining symbiotic associations between plants and beneficial microorganisms [32,61]. In legume-rhizobium nitrogen-fixing symbiosis, Ca^2+^ signaling is indispensable for both nodule formation and nitrogen fixation. Ca^2+^ oscillations in root hair cells are initiated upon recognition of rhizobial signals, leading to downstream responses that result in root hair deformation and initiation of infection thread formation [62]. Ca^2+^ waves subsequently propagate to nodule primordia, guiding organogenesis and nodule development [63].

Mycorrhizae are mutualistic symbioses established between most terrestrial plants and particular soil fungi. These interactions are sometimes ubiquitous, as one fungus can colonize different host plants, and some plants can form mycorrhizal associations with several fungi [64,65]. There are four types of mycorrhizas, among which arbuscular mycorrhiza (AM) are the most abundant and the most studied [65]. In mycorrhizal associations, Ca^2+^ signaling is involved in the recognition of fungal signals and subsequent establishment of a mutualistic symbiosis. Ca^2+^ oscillations in the plant root cells trigger downstream responses necessary for fungal colonization and nutrient exchange. To set up the intimate relation, plants react to AM fungal signals (lipochitooligosaccharides, short chain chitooligosaccharides) with Ca^2+^ spiking, the same response to Nod factors from rhizobia [33,66]. In fact, several plants defective in nodulation have also shown to be impaired in mycorrhization, indicating that certain elements of signal transduction are common to both symbiotic interactions [66,67].

### 4.1. Nodulating Rhizobia

In the work of Howieson [31], it was shown that external Ca^2+^ addition improved the attachment of several *Bradyrhizobium* strains to *Medicago* roots. The authors hypothesized that, as root and rhizobial cells possess a net negative charge in their surfaces, Ca^2+^ might assist in the linkage of free carboxyl groups in the peptidoglycan layer of rhizobia with organic acids on the root surface. However, this kind of electrostatic interaction does not reflect the specificity of the rhizobia–legumes relation.

A CaBP with demonstrated function is the calmodulin-like protein calsymin from the plant symbiotic microorganism *Rhizobium etli*. It presents three domains with predicted EF-hand motifs. The *casA* gene encoding calsymin is exclusively expressed during colonization and infection of *R. etli* on bean roots. Furthermore, mutation of *casA* affects the development of bacteroids during symbiosis and nitrogen fixation [68]. Previous studies have demonstrated the presence of high amounts of Ca^2+^ ions in nodules, which play a role in transporting fixed NH_4_^+^ to the cytoplasm or may interact with CDPK proteins expressed by plants in nodules [69]. Therefore, the secretion of CaBPs such as calsymin by rhizobia could function as an information transducer between the bacteria and the plant [44,70]. Homologs have also been found in the *R. leguminosarum* genome [71] (Table 1).

Another CaBP secreted by rhizobia is the nodulation protein NodO from *R. leguminosarum* and *Sinorhizobium* sp. BR816 [30,72]. NodO presents a β-roll motif, and it is secreted by the PrsD-PrsE Type I Secretion Systems (T1SS) [68]. It has been demonstrated that NodO inserts into liposome membranes and forms cation-selective channels and could therefore enhance the Ca^2+^ spiking that is observed in root hairs upon Nod factor binding [73]. Based on the experimental results, three possible functions of NodO have been established: (i) it may facilitate Nod factor uptake by the host; (ii) it can amplify the perceived Nod factor signal; or (iii) it may bypass the host’s Nod factor receptor altogether [74]. Nod factors initiate many developmental changes in the host plant during the onset of the nodulation process to generate the nodule primordium: root hair deformation, membrane depolarization, intracellular Ca^2+^ oscillations, and the initiation of cell division in the root cortex [75].

Rhizobial proteins secreted via the Type III secretion system (T3SS) can have beneficial, neutral, or detrimental effects on legume symbiosis [76]. The gene expression of the T3SS machinery is induced by flavonoids secreted by plants [77]. In *Bradyrhizobium diazoefficiens* USDA110, which can fix N_2_ in symbiosis with soybean and some varieties of mung bean, the T3SS is highly expressed in legume nodules [78], and it serves to deliver different effector proteins into the plant cells called Nodulation outer proteins, or Nop [76,79]. NopE1 and NopE2 are effector proteins that contain a Ca^2+^-dependent “metal ion-induced autocleavage” (MIIA) domain [61,80]. Several studies demonstrated that these proteins are transported from bacteria cells into the plant cell compartment of nodules, and that the Ca^2+^ concentration of this compartment allows the self-cleavage and the oligomerization of the resulted fragments, which are essential for host-specific legume-rhizobium symbiosis [61,80,81,82,83]. NopE homologs were found in several bradyrhizobia, but surprisingly, they were not identified in nearly any *B. elkanii* bacteria or in any *Sinorhizobium, Ensifer, Rhizobium*, or *Mesorhizobium* species [81]. Studies performed on an additional NopE homolog found in *Vibrio coralliilyticus*, the causative agent of coral bleaching, demonstrated that the MIIA domain of these proteins is intrinsically disordered and that the binding of Ca^2+^ induced the adoption of a secondary structure, with the consequent functionality of the proteins [84].

**Table 1 plants-12-03398-t001:** List of putative homologs of proteins involved in maintaining Calcium homeostasis and CaBP that were identified in the genomes of PGPR isolates from our laboratory groups.

	Organism	Function	Evidence and Reference	Protein Homology in *P. chlororaphis* subsp. *aurantiaca* SMMP3 Genome (GCA_018904775.1) ^1^	Protein Homology in *P. donghuensis* SVBP6 (CP129532.1) ^1^	Protein Homology in *Pseudomonas* sp. BP01 (GCF_022760795.1) ^1^	Protein Homology in *Methylobacterium* sp. 2A
**ATP Driven Transporters**							
YloB (NP389448.1)	*Bacillus subtilis* subsp. *subtilis* str. 168	P-type calcium transport ATPase	Heat resistance and early germination of spores. No effect on Ca^2+^ flux. Forms Ca^2+^ dependent phosphoenzyme intermediate at low ATP concentration in sporulating bacteria [85]	27.0% and 27.6% with MgtA (RS08220 and RS04100, respectively)	0	23.6% with KdpB (RS07175)	30.3% with MgtA and 33.17% with heavy metal translocating P-type ATPase (WP_160534356.1 and WP_116655330.1 respectively)
CaxP (YP816843)	*Streptococcus pneumonia* D39	Cation transporting P-ATPase	Protects cells against Ca^2+^ toxicity; is required for pathogenesis; chemical inhibition of CaxP is bacteriostatic at elevated Ca^2+^; affects Ca^2+^ flux [86]	30.1% and 29.3 with MgtA (RS08220 and RS04100, respectively)	25.7% homology with KdpB (RS09690)	0	30.8% with MgtA and 35.18% with heavy metal translocating P-type ATPase (WP_160534356.1 and WP_160536393.1, respectively)
PMA1 (WP_010872526.1)	*Synechocystis* sp. PCC6803	E1-E2 ATPase	Ca^2+^-dependent phosphorylated enzyme, inhibited by several specific inhibitors and induced by thaspigargin and ionophore A23187 [87]	0	0	0	35.2% with MgtA and 34.84% with heavy metal translocating P-type ATPase (WP_160534356.1 and WP_160536393.1, respectively)
PacL (BAA03906.1)	*Synechocystis elongatus* PCC7942	Ca^2+^ transporting P-ATPase	ATP-dependent Ca^2+^ uptake; plays no role in cell sensitivity to Ca^2+^ [88]	0	0	0	30.8% with MgtA and 39.05% with heavy metal translocating P-type ATPase (WP_160534356.1 and WP_160536393.1, respectively)
LMCA1 (CAC98919.1)	*Listeria monocytogenes*	Ca^2+^ transporting P-ATPase	Exchanges H+ for Ca^2+^ by ATP-dependent transport [89]	0	0	0	28.6% with MgtA and 25.51% with heavy metal translocating P-type ATPase (WP_160534356.1 and WP_160532634.1, respectively)
Lm0818 (NP464345.1)	*Listeria monocytogenes*	Ca^2+^ transporting P-ATPase	Structure resembles LMCA1 [90]	0	0	0	29.5% with MgtA and 25.96% with heavy metal translocating P-type ATPase (WP160534356.1 and WP_160536393.1, respectively)
PA2435 (NP252609.1)	*Pseudomonas aeruginosa* PA01	Putative cation-transporting P-type ATPase	Transposon mutant accumulates intracellular Ca^2+^; plays no role in cell tolerance to Ca^2+^ [91]	81.3% with cation translocating P-type ATPase (RS19670). 34.5% and 34.1% with heavy metal translocating P-type ATPases (RS18215 and RS20680, respectively)	35.6% and 35.8% homologies with copper-translocating P-type ATPases (RS12730 and RS26340, respectively). 29.6% with CcoS (RS18660)	36.2% and 34.7% homologies with heavy metal translocating P-type ATPases (RS20695 and RS23265, respectively)	48.0% and 46.83% with heavy metal translocating P-type ATPases (WP160532634.1 and WP_160536393.1, respectively)
PA3920 (CopA1, NP_252609.1)	*Pseudomonas aeruginosa* PA01	Putative metal transporting P-type ATPase	Plays role in Ca^2+^-induced swarming motility; transposon mutant accumulates intracellular Ca^2+^; plays no role in cell tolerance to Ca^2+^ [91]	76.7% and 34.2% with heavy metal translocating P-type ATPases (RS20680 and RS18215, respectively). 36.7% with a cation-translocating P-type ATPase (RS10725)	76.2% and 35% homologies with copper-translocating P-type ATPases (RS26340 and RS12730, respectively). 35.6% with CcoS (RS18660)	76.8% and 34.8% homologies with heavy metal translocating P-type ATPases (RS23265 and RS20695, respectively)	48.0% and 46.83% with heavy metal translocating P-type ATPases (WP160532634.1 and WP_160536393.1, respectively)
AtpD (NP418188.1)	*Escherichia coli* str. K-12 substr. MG1655	Beta subunit of F0-F1 ATP synthase	Δ*atpD* mutant is defective in Ca^2+^ efflux and has lower growth rate and ATP content at high Ca^2+^ [92]	84.6% with F0F1 ATP synthase subunit beta (RS19895)	85.0% with F0F1 ATP synthase subunit beta (RS02025)	84.% with F0F1 ATP synthase subunit beta (RS08435)	69.7% with F0F1 ATP synthase subunit beta and 29.24% with flagellar protein export ATPase FliI (WP160535066.1 and WP_161393307.1, respectively)
CtpE (AFP41926.1)	*Mycobacterium smegmatis* MC2-155	Metal-cation transporter P-type ATPase	Responsible for Ca^2+^ uptake. Disruption of *ctpE* resulted in a mutant with impaired growth under Ca^2+^-deficient conditions [93]	26.6% with MgtA (RS04100)	28.2% with CcoS (RS18660)	Between 25.4% and 27.3% with P-type ATPases (RS23265, RS00105, RS20695, RS07175)	28.6% with heavy metal translocating P-type ATPase and 27.45% with MgtA (WP160536393.1)
**Electrochemical potential driven transporters**							
ChaA (YP_489486.1)	*Escherichia coli* str. K-12 substr. W3110	Ca^2+^/H^+^ antiporter	Ca^2+^ efflux at alkaline pH [94]	30.1% with calcium:proton antiporter (RS20620)	0	0	39.6% with calcium:proton antiporter (WP_160533820.1)
PitB (AAC76023.1)	*Escherichia coli* str. K-12 substr. MG1655	Metal phosphate/H+ symporter	Performs Pi-dependent uptake of Ca^2+^ and Mg^2+^; Ca^2+^ uptake is inhibited by Mg^2+^ [95]	53.5% and 42.8% with inorganic phosphate transporters (RS25555 and RS12560, respectively)	53.1% and 42.0% with inorganic phosphate transporters (RS12245 and RS13850)	53.1% and 43.7% with inorganic phosphate transporter (RS14380 and RS06890)	43.3% and 40.7% with inorganic phosphate transporters (WP160536276.1 and WP_202131005.1, respectively)
Pit (O34436.2)	*Bacillus subtilis* subsp. *Subtilis* str. 168	Low affinity inorganic phosphate transporter	Performs Pi-dependent low affinity transport of Ca^2+^ [96]	38.0% and 33.8% with inorganic phosphate transporters (RS12560 and RS25555, respectively)	33.3% with inorganic phosphate transporter (RS12245). 36.2% with anion permease (RS13850)	33.3% and 36.7% with inorganic phosphate transporters (RS14380 and RS06890)	44.8% and 36.4% with inorganic phosphate transporters (WP202131005.1 and WP_160536276.1, respectively)
PA2092 (NP250782.1)	*Pseudomonas aeruginosa* PA01	Probable major facilitator superfamily (MFS) transporter	Transposon mutant accumulates intracellular Ca^2+^; plays no role in cell tolerance to Ca^2+^ [91]	Between 28% and 34% with 8 MFS transporters (RS17735, RS09905, RS07405, RS26135, RS26350, RS25660, RS07320, and RS27605)	Between 30.5% and 35.5% with 5 MFS transporters (RS10580, RS10255, RS12120, RS07855, and RS11115)	Between 28.8% and 38% with 6 MFS transporters (RS06440, RS10375, RS04620, RS02240, RS02735, and RS14175)	Between 21.3% and 35.9% with 7 MFS transporters (WP_202130650.1, WP_160532746.1, WP_202131106.1, WP_160533753.1, WP_161393615.1, WP_160533273.1, and WP_202130800.1)
**Channels**							
YetJ (O31539.1)	*Bacillus subtilis* subsp. *Subtilis* str. 168	pH sensitive Ca^2+^ leak channel	It has Ca^2+^ leak activity; performs two-phase Ca^2+^ influx regulated by pH [97]	0	0	0	32.5% with Bax inhibitor-1/YccA family protein (WP_160533491.1)
PA4614 (MscL, NP_253304.1)	*Pseudomonas aeruginosa* PA01	Conductance mechanosensitive channel	Transposon mutant accumulates intracellular Ca^2+^; plays role in Ca^2+^—induced swarming motility, but not in cell tolerance to Ca^2+^ [91]	86.0% with MscL (RS26905)	88.2% with MscL (RS02915)	90.5% with MscL (RS15690)	49.3% with MscL (WP_160532811.1)
**Regulators/Ca^2+^-binding proteins**							
PA0327 (CarP, NP_249018.1)	*Pseudomonas aeruginosa* PA01	Ca^2+^-regulated beta-propeller protein	Mutations in *carP* affected Ca^2+^ homeostasis, reducing the ability of *P. aeruginosa* to export excess Ca^2+^ [98]	63.8% and 39.2% with SdiA-regulated domain-containing proteins (RS18280 and RS18285, respectively)	68.2% and 40.7% with DNA-binding proteins (RS12660 and RS12655, respectively)	68.4% and 38.5% with SdiA-regulated domain-containing proteins (RS20635 and RS20630, respectively)	0
PA0320 (CarO, NP_249011.1)	*Pseudomonas aeruginosa* PA01	Ca^2+^-regulated OB-fold protein	Mutations in *carO* affected Ca^2+^ homeostasis, reducing the ability of *P. aeruginosa* to export excess Ca^2+^ [98]	58.1% with NirD/YgiW/YdeI family stress tolerance protein (RS21890)	62.4% with hypothetical protein (RS13100)	61.5% with NirD/YgiW/YdeI family stress tolerance protein (RS09335)	0
PA4107 (EfhP, NP_252796.1)	*Pseudomonas aeruginosa* PA01	Putative Ca^2+^-binding protein, with an EF-hand domain	The lack of EfhP abolished the ability to maintain intracellular Ca^2+^ homeostasis [99,100]	64.4% with EF-hand domain-containing protein (RS04190)	0	58.2% with hypothetical protein (RS10595)	0
LadS (NP_252663.1)	*Pseudomonas aeruginosa* PA01	Histidine kinase with 7 transmembrane and a Ca^2+^-binding DISMED2 domain	Triggers a Ca^2+^-induced switching between acute and chronic type of virulence, activating de Gac/Rsm cascade [101]	64.6% with the hybrid sensor histidine kinase/response regulator (RS26655)	65.4% with the hybrid sensor histidine kinase/response regulator (RS15480)	64.4% with ATP-binding protein (RS21750)	37.5% with ATP-binding protein (WP_202130978.1). 36.59% and 35.31% with response regulators (WP_160535701.1 and WP_161392897.1, respectively)
RapA2 (AUW48277.1)	*Rhizobium leguminosarum*	Ca^2+^-binding lectin	It possesses a cadherin-like β sheet structure that specifically recognizes capsular and extracellular exopolysaccharides [102]	27.7% with VCBS domain-containing protein (RS30095)	0	27.4% with VCBS domain-containing protein (RS12015)	0
AprA (NP_249940.1)	*Pseudomonas aeruginosa* PA01	Extracellular alkaline protease	Binds Ca^2+^ in EF-hand-like motifs, stabilizing its conformation and enzymatic activity [103]	65.0% and 63.2% with serralysin-family metalloproteases (RS05680 and RS01190, respectively)	64.7% with serine 3-dehydrogenase (RS06115)	0	0
LapF (NP_742967.1)	*Pseudomonas putida* KT2440	Surface adhesion protein	Ca^2+^ binding is necessary for correct folding at the periplasmic space. It has been shown that it participates in bacterial adhesion to seed and roots [104]	Between 30.1 and 33.9% with Ig-like domain containing proteins (RS29225, RS05900)	0	81.7% with the Ig-like containing protein (RS19225)	0
GspG (WP_000738789.1)	*Vibrio cholerae* RFB16	Type II secretion system major pseudopilin	It was demonstrated that altering the coordinating aspartates of the Ca^2+^ site in GspG dramatically impairs the functioning of the T2SS [105]	55.6% with the type II secretion system major pseudopilin GspG (RS16750)	56.5% with the type II secretion system GspG (RS04790)	55.8% with the type II secretion system major pseudopilin GspG (RS11585)	0
MxaF (SOR27906.1)	*Methylobacterium extorquens* TK0001	Methanol dehydrogenase, α subunit precursor	*mxaF* gene encodes a Ca^2+^-dependent MDH that contains Ca^2+^ in its active site and catalyzes methanol oxidation during growth on methanol [106]	24.6% with a glucose/quinate/shikimate family membrane-bound PQQ-dependent dehydrogenase (RS17145)	34.8% with a PQQ-dependent dehydrogenase, methanol/ethanol family (RS23825)	34.6% with the PQQ-dependent alcohol dehydrogenase PedH (RS02935)	88.96% with the methanol/ethanol family PQQ-dependent dehydrogenase (WP_202130689.1)
**Sensors**							
CarS (AAG06044.1)	*Pseudomonas aeruginosa* PA01	Ca^2+^-Regulated Sensor, part of the two-component system CarSR	Ca^2+^ binding induces the production of pyocyanin and pyoverdine and contributes to the regulation of the intracellular Ca^2+^ homeostasis and tolerance to elevated Ca^2+^ [98]	66.4% with sensor histidine kinase (RS07475)	65.7% with HAMP domain-containing protein (RS17265)	66.0% with sensor histidine kinase (RS10880)	37.4% and 32.5% with sensor histidine kinases (WP_202130576.1 and WP_160532461.1, respectively). 45.92% with ATP binding protein (WP_160535462.1). Between 30.6% and 33.7% with HAMP domain-containing sensor histidine kinase (WP_202130905.1 and WP_161392996.1, respectively)
CiaH (P0A4I6.1)	*Streptococcus mutans* UA140	Double-glycine-containing small peptide with a SD-domain shared by Ca^2+^-binding proteins	CiaX responds negatively to Ca^2+^. Cation mediated cell functions and biofilm production [107]	0	0	0	32.6% with ATP binding protein (WP_160532632.1). 29.3% with a response regulator (WP_160535701.1). 27.6% with HAMP domain-containing sensor histidine kinase (WP_202130905.1)
CvsS (NP_793163.1)	*Pseudomonas syringae* pv. tomato DC3000	Ca^2+^-Regulated Sensor, part of the two-component system CvsSR	Virulence-associated sensor that is induced by Ca^2+^ in vitro and in planta, regulating T3SS and AlgU [108]	73.9% with sensor histidine kinase (RS07475)	73.5% with the HAMP domain-containing protein (RS17265)	71.0% with the sensor histidine kinase (RS10880)	28.48% with sensor histidine kinase (WP_160532461.1)
PhoP (NP_249870.1)	*Pseudomonas aeruginosa* PA01	Ca2+-Regulated Sensor, part of the two-component system PhoPQ	Negatively affected by Ca^2+^. PhoPQ system enhances resistance to antimicrobial peptides and is responsible of lipid A modifications [109]	67.2% with ATP-binding protein (RS12960)	67.9% with two-component sensor histidine kinase (RS07415)	67.4% with ATP-binding protein (RS01970)	Between 46.6% and 33.8% with 9 response regulator transcription factors (WP_202130544.1, WP_202130575.1, WP_202131087.1, WP_160535441.1, WP_246730681.1, WP_202130521.1, WP_160533116.1, and WP_160534746.1)

In gray, we highlight the homologies with more than 50% similarity. We did not find any homolog of these proteins: Cda (AAC41526.1); bacteriorhodopsin (YP_001689404.1); Ca^2+^/H^+^ antiporters ApCAX (BAD08687.1), SynCAX (P74072.1) and ChaA (NP_388673.1); LmrP (NP268322.1); Calsymin (AAG21376.1); CabD (3AKA_A); NodO (CAK10399.1); CcbP (WP_099068791.1); Protein S (WP_011555392.1). ^1^ For each homology search, the references in parenthesis show the Locus Tag of each protein obtained from the *Pseudomonas* database (www.pseudomonas.com, accessed on 5 September 2023).

Other CaBPs involved in the nodulation process are adhesins [110]. One of them is the well-studied rhicadhesin from *Rhizobium leguminosarum* biovar viciae 248, which mediates the first step in the attachment of bacterial cells to plant root hair tips under neutral or alkaline conditions [111]. However, the gene encoding this protein is still unknown. It has been demonstrated that rhicadhesin can bind Ca^2+^ and that this interaction is necessary for the adhesin activity [112]. This protein is able to inhibit the attachment of other rhizobia, such as *Agrobaterium, Bradyrhizobium,* and different *Rhizobium* species, to the roots of several plants, including legumes, non-legumes, and monocots [110]. In fact, rhicadhesin-mediated attachment is common in the entire Rhizobiaceae family but apparently is not involved in the adhesion of other plant growth-promoting genera, such as *Azospirillum* and *Pseudomonas* [110]. Apparently, Ca^2+^ anchors rhicadhesin to the bacterial cell [112], as it was shown that under Ca^2+^-limited conditions the bacterial adhesion to roots was reduced [112,113]. The cation is not involved in the interaction between rhicadhesin and the rhicadhesin-receptor found in plant cells [114]. This model could not be specifically demonstrated, given that rhizobia mutants lacking rhicadhesin expression were not obtained yet. Nevertheless, chromosomal virulence mutants of *A. tumefaciens (chvB)* impaired in attachment ability to plant cells had no active rhicadhesin, and the wildtype phenotype could be restored by the external addition of purified rhicadhesin [115,116].

Other Ca^2+^-binding adhesins include the *Rhizobium*-adhering proteins (Rap) [117]. RapA proteins are secreted CaBPs produced only by some *R. etlii* and *R. leguminosarum* representatives. RapA1 generates the bacterial cell agglutination and improves root colonization levels, although nodulation is not enhanced [117,118]. RapA2 has a lectin function as it binds to acidic exopolysaccharides, thus suggesting a role in the development of the biofilm matrix of rhizobia [119]. RapA1 and RapA2 possess a cadherin-like domain (CHDL), which was described in a family of eukaryotic CaBPs [119]. CHDL domains are responsible for the initiation and maintenance of cell-cell contacts through a Ca^2+^-dependent mechanism by protein-protein interaction or carbohydrate binding [120]. Adhesins are ubiquitous in the bacterial kingdom, and in silico analyses suggest that putative RapA2 homologs (VCBS domain-containing proteins) are present in the genomes of *P. chlororaphis* SMMP3 and *Pseudomonas* sp. BP01 isolates (Table 1).

Thus, in conclusion, Ca^2+^ plays an essential role in rhizobia-plant interaction. Given that growing root hairs and pollen tubes exhibit tip-focused gradients of Ca^2+^ [75], all the aforementioned evidence highlights that the environmental presence of Ca^2+^ favors the root colonization and nodulation processes with a sum of different mechanisms.

### 4.2. Mycorrhizal Fungi

Little is known about the role of Ca^2+^ in the mycorrhizal fungal cells, although in other fungi, the CaBP caleosin was associated with conidial germination, lipid storage, and infection ability [121,122,123]. Using a transcriptomic approach, Tisserant et al. showed an up-regulation of two CaBPs in the intraradical mycelium of *Glomus intraradices*, an AM fungus, although they did not discuss this upregulation [124]. The up-regulation of *G. intraradices* Ca^2+^-related genes has been reported during both the early (appressorium/hyphopodium) and late (arbuscule) stages of symbiotic interactions with different host roots, strongly suggesting the involvement of Ca^2+^ in fungal processes leading to root colonization [32,125]. In several transcriptomic analyses, Li et al. found that two Ca^2+^-related genes, coding for an ATPase involved in Ca^2+^ transport and a CaBP, might be important for the AM fungus *Rhizophagus irregularis* response to the plant hormone strigolactone [126]. In the same study, they determined that these genes were acquired by horizontal gene transfer from plants, indicating that the acquisition of genes encoding CaBPs might have facilitated *R. irregularis*-plants symbiosis [126].

## 5. Ca^2+^ Signaling in PGPR-Plant Interactions

Diverse calmodulin-like proteins have been reported in several species of actinomycetes [127,128]. Calerythrin was the first “true” EF-hand CaBP (with at least one pair of interacting EF-hands, Figure 1A) described in prokaryotes, specifically in *Saccharopolyspora erythraea* (formerly *Streptomyces erythreus*) [129]. Other calmodulin-like proteins have been found in *Streptomyces* species, such as *S. ambofaciens* and *S. coelicolor*, which were named *cabB*, *cabC*, *cabD*, and *cabE* [59,128] (Table 1). Ca^2+^ was reported to be involved in many processes in these Gram-positive bacterial genera, such as spore germination and aerial mycelium formation [130,131]. Apparently, calerythrin and CabB only serve as Ca^2+^ buffers and/or transporters that modify the spatiotemporal [Ca^2+^] in the bacterial cell without any direct implication in these morphological processes [132,133]. However, the role of CabC in the regulation of spore germination and aerial hyphal growth has been clearly demonstrated [134]. The functions of CabD and CabE still remain unclear, despite some bioinformatics approaches that have made some suggestions about this topic [135].

Extracellular [Ca^2+^]_ex_ plays a direct or indirect signaling role in many of the bacterial protein secretion systems. Bacteria use at least eight different strategies to secrete proteins, termed T1SS to T8SS [136]. Some extracellular proteins that depend on Ca^2+^ to fold and function correctly are responsible for the biocontrol activities of PGPR strains. A representative of this group, the metalloprotease AprA, was experimentally demonstrated to be Ca^2+^-dependent [103]. It possesses a C-terminal domain containing a nonapeptide Gly- and Asp-rich repeat with a β-roll motif characteristic of the repeats in toxin (RTX) Ca^2+^-binding domains. Ca^2+^-binding RTX proteins are secreted by the T1SS [137] and they acquire their native structure with a Ca^2+^-dependent folding during secretion [52]. AprA is widely distributed inside the *Pseudomonas* genus, accomplishing several roles in different species (Table 1; Figure 1C); it is involved in the biocontrol of the root-knot nematode *Meloidogyne incognita* performed by *Pseudomonas protegens* CHA0 in tomato and soybean [138], and prevents the predation of *P. protegens* CHA0 by several protists in the rhizosphere, improving its colonization competence [139]. In the entomopathogenic bacteria *Pseudomonas entomophila,* AprA showed insecticidal activity against the bean bug *Riptortus pedestris* via the suppression of host cellular and humoral innate immune responses [140], and in the phytopathogenic bacteria *Pseudomonas syringae* pv. *tomato* DC3000, AprA is responsible for the degradation of flagellin subunits, which are strong inducers of innate immune responses in plants, thus improving its virulence in tomato and *Arabidopsis thaliana* [141].

Immune evasion, a mechanism used by pathogens to infect plants and mammals, could also be used by non-symbiotic PGPR strains. *Pseudomonas brassicacearum*, an excellent root-colonizer with biocontrol potential [142,143], employs a phase variation strategy to evade the plant immune system and reach high root-colonization levels [144]. It was demonstrated that protease and lipase activities are only present in the phase I cells, and that this phenotype is due to the expression of the operon containing the *aprA* gene [145]. These phase I cells are localized at the basal part of roots, whereas phase II cells are present on young roots and root tips [144], suggesting that phase I cells are the first ones to contact roots and employ AprA to reduce their antigenic potential [145]. These results support the idea that phase variation plays a role in immune evasion. Although it is yet to be demonstrated, the folding of AprA in the rhizospheric environment could be accomplished since Ca^2+^ is a relatively abundant element in soils, with average concentrations ranging from 7 to 24 mg/g of soil [146], with weathered limestone and the weathering of certain primary minerals being the main sources [147,148]. Hence, the folding of AprA in the rhizospheric environment could be accomplished, although it is yet not experimentally demonstrated.

Other lytic enzymes are secreted via the T2SS [149]. In contrast with AprA, T2SS substrates must be folded in the periplasm for recognition by the secretion machine. T2SS mutants accumulate otherwise secreted proteins in the periplasm [150]. Several plant-pathogenic bacteria secrete their virulence factors via this secretion system [151]. It was reported that phosphatases involved in the P-uptake of *P. putida* KT2440 are also released via this secretion system [152]. Inorganic phosphorus solubilization (by organic acid production) or phosphate releasing (by phosphatases) are rewarding plant growth-promoting traits that are sought for the development of biofertilizers [153]. The sophisticated multiprotein machinery T2SS is widespread in Proteobacteria and contains 11–15 different proteins, among which is the major pseudopilin (GspG in *Vibrio cholerae*). Several crystal structures from different bacteria revealed that a Ca^2+^ ion is bound at the same site in pseudopilin. Altering the coordinating aspartates of the Ca^2+^ site in GspG dramatically impairs the functioning of the T2SS [105]. This secretion system is commonly found in Gram negative bacteria; in fact, we found homologues of GspG in our *Pseudomonas* isolates (Table 1). In conclusion, the correct assembly of the T2SS responsible for the secretion of a wide spectrum of lytic enzymes involved in biofertilization or pathogenesis requires Ca^2+^.

*Bacillus* genera produce extracellular α-amylases that allow them to process several starch sources [154]. Proteins from the α-amylase family contain an EF-hand-like motif with a demonstrated Ca^2+^ binding activity [155,156,157,158] (Figure 1B). In this protein family, Ca^2+^ is essential for their catalytic activities because its binding contributes to their correct structure adoption and thermostability [156,159]. These enzymes are important biological products for clinical and industry practices [160,161]. To improve the process efficiency and reduce industry costs, some approaches seek Ca^2+^ independence [162]. Despite its well-established role as PGPR [163], it has been recently suggested that the presence of members of the *Bacillus subtilis* complex in starchy storage plant organs promotes the development of rot diseases by opportunistic pathogens. For example, *Pantoea ananatis* and *P. agglomerans* can consume the fast assimilable carbon sources that are generated after the starch degradation by amylases in onion bulbs or potato tubers, respectively. Under such favorable conditions, phytopathogenicity develops [164].

The ability to establish and colonize plant tissues, particularly roots, is intimately related to robust biofilm formation [165,166]. Biofilm matrixes are composed of extracellular polymeric substances (EPS), which consist of exopolysaccharides, nucleic acids (eDNA and eRNA), proteins, lipids, and other biomolecules [167]. Due to the ability of Ca^2+^ to interact with surface proteins and form ionic bridges between negatively charged macromolecules (such as the EPS alginate, the eDNA, and the eRNA), this cation enhances cell aggregation and strengthens biofilm matrixes [168,169,170]. Additionally, like in rhizobia, adhesins also play an important role in bacterial cell adhesion to plant surfaces (Figure 1E). The natural soil isolate *Pseudomonas brassicacearum* WCS365 contains LapA, a large protein loosely associated with the bacterial surface that is involved in the cell attachment to surfaces during the early stages of biofilm formation. LapA determines the transition from reversible to irreversible attachment [171,172]. Another large repetitive protein found in the bacterial surface, LapF, was described as the main factor for the development of a mature biofilm in *P. putida* KT2440 [173]. It was demonstrated that LapA and LapF are essential for the correct bacterization of maize seeds and the root colonization by *P. putida* KT2440 [104,173,174]. We identified a homolog of LapF in the genome of *Pseudomonas* sp. BP01 (Table 1). LapF contains a C-terminal domain typical of the proteins secreted via T1SS, with a β-roll Ca^2+^ binding motif [175] (Figure 1E). It was shown that Ca^2+^ is involved in the multimerization of LapF, improving the ability of *P. putida* KT2440 to form biofilms. In contrast, in *P. fluorescens* Pf0-1, which does not contain a homolog of LapF in its genome and only synthesizes LapA, Ca^2+^ negatively regulates its ability to form biofilms [176]. 

Bioinformatic predictions have shown that LapA contains a Calx-β domain, which is involved in Ca^2+^ binding in eukaryotic Na^+^/Ca^2+^ exchangers [177] and typical RTX repetition sequences in its C-terminus. Deletion of Calx-β in LapA did not affect the adhesion and biofilm formation of *P. fluorescens* Pf0-1 on a hydrophobic surface [153]; however, deletion of the C-terminus reduced biofilm formation on both hydrophobic and hydrophilic surfaces [178]. RTX repeats bind Ca^2+^ with high affinity in other adhesins [179], but this is not experimentally demonstrated for LapA. However, it was shown that Ca^2+^ is essential for the correct folding and activity of LapG, a periplasmic cysteine protease, which acts on the N-terminus of LapA, causing its detachment from the bacterial surface [176,178]. In conclusion, Ca^2+^ possibly contributes to the colonization process, improving the bacterial competition for niche establishment and the attachment to plant tissues.

Myxobacteria are spore-forming δ-Proteobacteria that develop fruiting bodies where the spores are conserved together until the optimal conditions for growing are present in the environment. Then, the germinated spores can re-establish their social behavior [180]. These microorganisms are important epibiotic soil predators because they secrete hydrolytic enzymes and diverse secondary metabolites that affect microbial growth [181]. Due to this characteristic, some myxobacteria isolates have been studied for the biocontrol of several bacterial and fungal phytopathogens, such as *Ralstonia solanacearum* in tomato [182], *Rhizoctonia solani* in pine [183], *Fusarium* wilt [184,185], *Botrytis cinerea*, *Colletotrichum acutatum*, and *Pyricularia grisea* [186], and *Phytophthora infestans*, the causative agent of potato late blight [187].

The myxobacteria *Myxococcus xanthus* contains a stress-induced protein known as protein S, a monomeric Ca^2+^-binding two-domain protein that is a member of the βɣ-crystallin superfamily [188]. The two domains show a high structural similarity, and each one contains two Ca^2+^-binding motifs from the Greek-key family [189] (Figure 1D). The monomeric protein S polymerizes in a Ca^2+^-dependent manner during spore development to form an extremely stable cuticula that allows myxospores to endure cryptobiosis over long periods of time [188]. Hence, Ca^2+^ plays an essential role in myxobacteria, because the assembly of this protein in the myxospores’ surface occurs in the presence of Ca^2+^ [190]. As expected, we did not find a homolog of protein S in the genomes of our *Pseudomonas* or *Methylobacterium* isolates that do not form spores (Table 1).

Cyanobacteria is another PGPR group in which Ca^2+^ plays an important role [191]. This ion is involved in heterocyst differentiation, a process necessary for biological N_2_ fixation in an oxygenic environment [192]. In the cyanobacterium *Anabaena* sp. strain PCC 7120, the Ca^2+^-binding protein CcbP is responsible for the formation of those specialized nitrogen-fixing cells. CcbP possesses two Ca^2+^-binding sites and adopts a new folding status when the cation is bound. Site I consists of an α-turn-β region unreported previously, whereas site II resembles a classical EF-hand motif. Ca^2+^ dependence was demonstrated when a mutation of Asp38 at the stronger Ca^2+^-binding site abolished its ability to regulate the heterocyst differentiation [193]. Although the importance of Ca^2+^ for heterocyst formation is clear, the signaling mechanisms remain unknown. Physiological and biochemical studies indicate that a low nitrogen/high carbon signal triggers high intracellular [Ca^2+^] that are detected by a Ca^2+^ sensor with two EF-hand domains (CSE), a protein that is found almost exclusively in filamentous cyanobacteria [194]. The CSE-encoding gene is downregulated during nitrogen limitation and upregulated during CO_2_ limitation, the conditions that induce heterocyst differentiation [195]. Finally, genomic analyses from different cyanobacteria (*Synechocystis* sp. PCC6803, *Anabaena* sp. PCC7120, and *Thermosynechococcus elongatus* BP-1) revealed the presence of a single Ca^2+^/H^+^ antiporter in their genomes. Biochemical studies of those proteins demonstrated that they are involved in the salt tolerance of those microorganisms [196] (Table 1). We did not identify CcbP homologs in *Pseudomonas* or *Methylobacterium* isolates.

It is well documented that plant leaves release methanol and experimental evidence suggests that most of the methanol produced inside leaves is emitted primarily through stomata [197]. Different microbial species can utilize the methanol synthesized by plants as a carbon and energy source. Methanol conversion to formaldehyde is catalyzed by different methanol dehydrogenases (MDHs) systems. The electrons released from this reaction are then passed through cytochrome (Cyt) to the end of the electron transfer chain, resulting in the generation of one ATP per methanol molecule [198]. The formaldehyde is then oxidized to CO_2_ in the cytoplasm in a pathway dependent on the cofactor tetrahydromethanopterin, or it is assimilated into cell biomass via the serine cycle. Gram-positive bacteria harbor a NAD(P)-dependent MDH in their cytoplasm while many Gram-negative methylotrophic taxa harbor Ca^2+^-dependent MxaFI type and lanthanide-containing XoxF type MDHs [106]. Core enzymes involved in methylotrophy in *M. extorquens,* the most studied representative of the facultative methylotrophs, have been characterized in detail [199]. Methanol is first oxidized to formaldehyde by a periplasmic MDH that is an heterotetramer (α2β2) composed of two large catalytic subunits (α, MxaF) and two small subunits (β, MxaI) [200]. The large subunits represent the catalytic part of the enzyme carrying one pyrroloquinoline quinone (PQQ) and one Ca^2+^ atom per subunit. The PQQ prosthetic group has the role of capturing electrons from methanol oxidation and passing them to Cyt. MxaFI-MDHs bind Ca^2+^ as a cofactor that assists PQQ in catalysis. The function of the small subunits, which tightly wrap against the large subunits, is elusive. The crystal structures of MxaFI-MDHs from several terrestrial species that use Ca^2+^ as a cofactor have been resolved (revised in [201]), and more recently, those from a marine methylotrophic bacterium, *Methylophaga aminisulfidivorans* MPT [202], where Ca^2+^ is replaced by Mg^2+^. Zhao reported that MDH activity increased significantly with the increasing concentration of Ca^2+^, approaching saturation at 200 mM Ca^2+^ [203]. The effect of Ca^2+^ on the activation of MDH was time dependent and Ca^2+^ specific and was due to binding of the metal ions to the enzyme. The activation of MDH by Ca^2+^ occurred concurrently with a conformational change. In addition, exogenously-bound Ca^2+^ destabilized MDH.

## 6. Effects of Calcium on the Growth of *Pseudomonas* and *Methylobacterium* Isolates

Throughout the course of evolution, cells have developed a Ca^2+^ toolkit, which enables them to respond to various stimuli while preserving their integrity. With the advantage of having sequenced the genomes of the *Pseudomonas* and *Methylobacterium* isolates characterized in our labs [12,13,14,15], we conducted a bioinformatic exploration to identify homologues of the Ca^2+^ channels, exchangers, pumps, and CaBPs identified in other microorganisms that are listed in Table 1 (highlighted in gray are those proteins detected in our isolates that share more than 50% sequence homology) and shown in Figure 2.

Among these proteins, we find the Beta subunit of F0-F1 ATP synthase in the four isolates. In addition, we identified different P-type ATPases that translocate cations, heavy metals, copper, potassium, or magnesium. We detected a Ca^2+^:H^+^ antiporter in *P. chlororaphis* SMMP3 and *Methylobacterium* sp. 2A, while the four isolates present inorganic phosphate transporters and major facilitator superfamily (MFS) transporters. We also identified mechanosensitive channels (MscL) in the four isolates and a Bax inhibitor-1/YccA family protein in *Methylobacterium* that shares low (32%) homology with a pH-sensitive Ca^2+^ leak channel. The search for Regulators/Ca^2+^-binding proteins showed that the *Pseudomonas* isolates, *P. chlororaphis* SMMP3 and *Pseudomonas* sp. BP01, harbor SdiA-regulated domain-containing proteins and NirD/YgiW/YdeI family stress tolerance proteins that are homologues of CarP and CarO, respectively, and *P. chlororaphis* SMMP3 presents a putative CaBPs with an EF-hand domain (Table 1). On the other hand, *Methylobacterium* harbors the MDH gene MxaF, but this protein is not present in the *Pseudomonas* isolates, although proteins from the family of PQQ dehydrogenases with low homology were encountered in them (Table 1).

To test if these isolates could maintain Ca^2+^ homeostasis in vitro, we cultured them in LBNS medium containing increasing [Ca^2+^]. These preliminary results show that excessively elevated [Ca^2+^]_ex_ (≥50 mM) exerted detrimental effects on their growth, viability, and colony size, as observed in Figure 3. However, cells developed quite well in media containing up to 10 mM Ca^2+^, and this tolerance could be linked with the detection of putative ATP-driven transporters and antiporters encoded in their genomes (Table 1, Figure 2). The addition of Ca^2+^ caused a significant reduction in the diameter of the colonies of our isolates compared to the control condition, and this effect was more evident with high [Ca^2+^]_ex_ in the media (Figure 3C). However, we observed different levels of Ca^2+^ tolerance among our isolates; for example, *P. donghuensis* SVBP6 was the most affected by high [Ca^2+^]_ex_ in solid and liquid media (Figure 3A,C). This observation is correlated with the fact that Ca^2+^-related proteins were not abundant within this strain (Table 1). In *P. chlororaphis* SMMP3, this reduction level was only noticeable when exposed to 100 mM Ca^2+^ (Figure 3C). Conversely, *Pseudomonas* sp. BP01 exhibited a reduction of less than 20% in the colony diameter up to [Ca^2+^]_ex_ = 50 mM and a reduction of less than 40% at the highest [Ca^2+^] (Figure 3C). In the case of *Methylobacterium*, high Ca^2+^ exerted a strong inhibition in growth and colony diameter (Figure 3B,C). This last effect was also observed in the presence of EGTA, although bacterial growth in the presence of the chelator was similar to controls (Figure 3B). These simple approaches suggest that there are differences in the Ca^2+^ toolkit and in the tolerance to high Ca^2+^ among PGPR strains, even from the same genus. As all the strains evaluated came from soil or rhizosphere samples and showed PGPR properties *in vitro*, these differences could play an important role in their interaction with plants and maybe should be considered for bioinput development.

## 7. Conclusions

The results compiled in this review demonstrate the crucial significance of Ca^2+^ in prokaryotes and highlight several topics related to its role in regulating protein function and cellular signaling networks in different physiological processes within PGPRs. However, a deeper exploration is required to unveil the role of this cation within the context of PGPR interactions with plants and other soil microorganisms. Most of the proteins reported to have demonstrated Ca^2+^ binding activity or function in prokaryotes were identified in human pathogenic bacteria. We used these proteins as a query for the in silico search of homologues in our PGPR isolates (Table 1), keeping in mind that the rise of intracellular Ca^2+^ levels is a typical response in both pathogenic and mutually beneficial plant/microbe interactions. As observed in Table 1 and Figure 2 and Figure 3, different bacterial species exhibit a variety of Ca^2+^ influx and extrusion systems and adopt several strategies in the course of evolution to proliferate in their ecological niches, employing to this end specific CaBP proteins. Recognizing its relevance, the prospect of modulating Ca^2+^ concentrations in bioformulations, soils, or rhizospheres emerges as an intriguing tool for enhancing the efficacy of microbiological candidates in the development of agricultural bioinputs.

## Figures and Tables

**Figure 2 plants-12-03398-f002:**
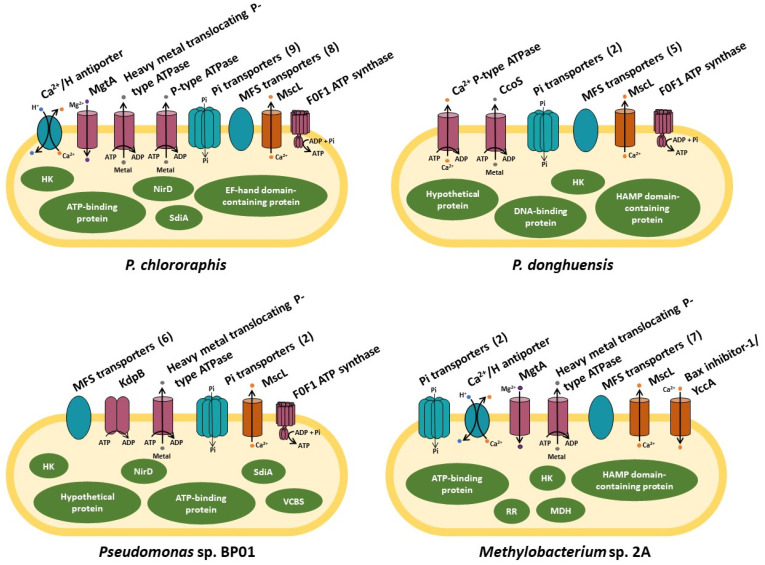
Proteins involved in maintaining Calcium homeostasis and CaBP identified in the genomes of *P. chlororaphis* subsp. *aurantiaca* SMMP3, *P. donghuensis* SVBP6, *Pseudomonas* sp. BP01, and *Methylobacterium* sp. 2A. The number of transporters found in each PGPR genome is indicated between brackets. ATP-driven transporters are colored in pink, electrochemical potential-driven transporters in light blue, channels in orange, and proteins in green. This figure was created from the information detailed in Table 1.

**Figure 3 plants-12-03398-f003:**
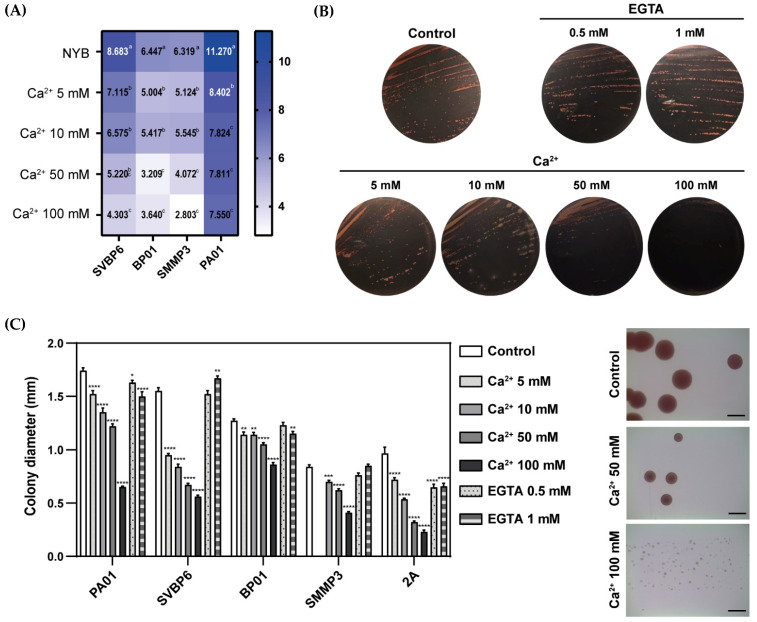
Increasing [Ca^2+^]_ex_ affected the growth and colony size of *Pseudomonas chlororaphis* subsp. *aurantiaca* SMMP3, *P. donghuensis* SVBP6, *Pseudomonas* sp. BP01, and *Methylobacterium* sp. 2A. The four isolates were grown on nutrient broth (NYB, liquid) or nutrient agar (NA solid) for *Pseudomonas* or LBNS media for *Methylobacterium* sp. 2A, all supplemented or not (controls) with 5, 10, 50, 100 mM Ca^2+^ or 0.5 and 1 mM EGTA. (**A**) Heatmap with multiple comparisons of the growth rate constants in liquid NYB media with increasing [Ca^2+^]_ex_ (µ = [(log_10_N–log_10_N_0_) 2.303)/(t–t_0_)]) of the *Pseudomonas* isolates and of the type strain *P. aeruginosa* PA01 performed. The OD_600_ values were measured in a microplate reader every 1 h for 20 h at 28 °C and 200 rpm. PA01 was included as a positive control of a well-studied *Pseudomonas* with a high [Ca^2+^]_ex_ tolerance [204]. Different letters indicate significant differences. (**B**) *Methylobacterium* sp. 2A was grown on solid LBNS with increasing [Ca^2+^]_ex_ and EGTA concentrations. Plates were photographed after 5 days of incubation at 28 °C. (**C**) Analysis of the effect of the addition of [Ca^2+^]_ex_ in solid media on the colony diameter for each isolate in the assayed concentrations (left panel). Asterisks indicate significant differences between each treatment and the control without Ca^2+^. Statistical analysis was performed by two-way ANOVA followed by Tukey’s HSD test (* *p* < 0.01, ** *p* < 0.001, *** *p* < 0.0001, **** *p* < 0.00001). Plates were photographed with a magnifying glass, and colony diameter (mm) was measured through ImageJ [205]. Representative pictures of *Methylobacterium* sp. 2A colonies in control, 50 mM, and 100 mM Ca^2+^ conditions are shown (right panel). The scale bars in the pictures represent 1 mm.

## Data Availability

Not applicable.

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
