# Peer review of "Unveiling the Secrets of Calcium-Dependent Proteins in Plant Growth-Promoting Rhizobacteria: An Abundance of Discoveries Awaits"

_plants, 2023, doi:10.3390/plants12193398_

Round 1

Reviewer 1 Report

This paper presents a comprehensive review and some new information on the role of calcium, particularly with respect to beneficial plant-microbe interactions and the biology of plant-associated microorganisms. Overall the paper is well written and will be useful to researchers in the field. I have only some minor comments:

- Since the focus is mostly on the microbial part of the association, perhaps sections 2 and 3 can be reduced.

- Also, I detected some redundances between the introduction and the different sections that could be polished.

- The results presented in Figure 3, while potentially interesting, seem rather preliminary at this point, and they divert the attention from the main aspects of the review. I would recommend leaving those results for a future, more detailed analysis of the influence of calcium on the biology of the different species/isolates.

- The section on pyocianin (lines 436-446) seem a bit out of place. I would directly go to explain the role of EPS an their potential cation-trapping activity.

Author Response

Dear Editor,

A new version of our manuscript Plants 2622561: “Unveiling the secrets of Calcium-dependent proteins in Plant-Growth Promoting Rhizobacteria: An abundance of discoveries awaits” with the corrections highlighted is now submitted. We hope you find this new version suitable for publication in this prestigious journal.  We thank the reviewers for their valuable comments, listed is the response to their comments and suggestions.

Reviewer 1:

This paper presents a comprehensive review and some new information on the role of calcium, particularly with respect to beneficial plant-microbe interactions and the biology of plant-associated microorganisms. Overall, the paper is well written and will be useful to researchers in the field. I have only some minor comments:

- Since the focus is mostly on the microbial part of the association, perhaps sections 2 and 3 can be reduced.

- Also, I detected some redundances between the introduction and the different sections that could be polished.

- The results presented in Figure 3, while potentially interesting, seem rather preliminary at this point, and they divert the attention from the main aspects of the review. I would recommend leaving those results for a future, more detailed analysis of the influence of calcium on the biology of the different species/isolates.

- The section on pyocianin (lines 436-446) seem a bit out of place. I would directly go to explain the role of EPS and their potential cation-trapping activity.

RTR: We thank Reviewer 1 for the valuable comments, we merged sections 2 and 3 to reduce them and eliminated the section on pyocyanin. Regarding the results presented in Figure 3 (section 6) we agree that these results are preliminary and this we stated in the text. However, we consider that using a simple approach we could highlight that the strains tested present different tolerance to high Calcium concentrations which correlates with the presence or not of proteins related to Ca2+ in their genomes. For this reason, we consider that it is worth maintaining this section.

Reviewer 2: The manuscript provides an in-depth discussion on the significance of Ca2+ in prokaryotes, particularly focusing on its role in regulating physiological processes within PGPR (Plant Growth-Promoting Rhizobacteria). The author acknowledges that most of the current information is derived from studies on human pathogenic bacteria, and emphasize the need for further exploration, specifically within the context of PGPR interactions with plants and other soil microorganisms. Additionally, the authors highlight the diversity of Ca2+ influx and extrusion systems among different bacterial species, as well as the adoption of specific CaBP (Calcium-Binding Protein) proteins during evolution to thrive in their ecological niches. The potential of modulating Ca2+ concentrations in bioformulations, soils, or rhizospheres is also suggested as a promising tool to enhance the efficacy of microbiological candidates in agricultural bio inputs.

Overall, this manuscript provides valuable insights into the functional role of Ca2+ in prokaryotes and proposes an intriguing application for manipulating Ca2+ concentrations in agricultural bio inputs. The study appears to be significant and offers potential implications for future research in related fields. However, to improve the manuscript, I would like to suggest the following points:

  1. While the discussion on Ca2+ regulation in human pathogenic bacteria is a good starting point, it would be beneficial to further elaborate on the mechanisms of Ca2+ regulation in other prokaryotes. Collecting and analyzing relevant literature could help to broaden the understanding of this field.

  1. Attention should be given to grammar errors and improving sentence structure to enhance the overall readability and flow of the manuscript.

RTR: We thank Reviewer 2 for the comments and appreciate the suggestions. We searched and compiled the research performed in other microorganisms focusing on PGPRs. Most of the proteins reported to have a demonstrated Ca2+ binding activity or function in prokaryotes were identified in human pathogenic bacteria. We used these proteins as query to find homologues in our PGPR isolates (Table 1) having in mind that the rise of intracellular Ca2+ levels is a typical response in both pathogenic and mutually beneficial plant/microbe interactions. In addition, we performed simple experiments to evaluate calcium tolerance in these PGPRs with the hope of contributing to a vacant area of research.

We improved the grammar and spelling of the manuscript.

Reviewer 2 Report

The manuscript provides an in-depth discussion on the significance of Ca2+ in prokaryotes, particularly focusing on its role in regulating physiological processes within PGPR (Plant Growth-Promoting Rhizobacteria). The author acknowledges that most of the current information is derived from studies on human pathogenic bacteria, and emphasize the need for further exploration, specifically within the context of PGPR interactions with plants and other soil microorganisms. Additionally, the authors highlight the diversity of Ca2+ influx and extrusion systems among different bacterial species, as well as the adoption of specific CaBP (Calcium-Binding Protein) proteins during evolution to thrive in their ecological niches. The potential of modulating Ca2+ concentrations in bioformulations, soils, or rhizospheres is also suggested as a promising tool to enhance the efficacy of microbiological candidates in agricultural bio inputs.

Overall, this manuscript provides valuable insights into the functional role of Ca2+ in prokaryotes and proposes an intriguing application for manipulating Ca2+ concentrations in agricultural bio inputs. The study appears to be significant and offers potential implications for future research in related fields. However, to improve the manuscript, I would like to suggest the following points:

1. While the discussion on Ca2+ regulation in human pathogenic bacteria is a good starting point, it would be beneficial to further elaborate on the mechanisms of Ca2+ regulation in other prokaryotes. Collecting and analyzing relevant literature could help to broaden the understanding of this field.

2. Attention should be given to grammar errors and improving sentence structure to enhance the overall readability and flow of the manuscript.

Author Response

(The authors gave the same response as above.)
